# Exploring the Implications of COVID-19 on Food Security and Coping Strategies among Urban Indigenous Peoples in Saskatchewan, Canada

**DOI:** 10.3390/nu15194278

**Published:** 2023-10-07

**Authors:** Mojtaba Shafiee, Ginny Lane, Michael Szafron, Katherine Hillier, Punam Pahwa, Hassan Vatanparast

**Affiliations:** 1College of Pharmacy and Nutrition, University of Saskatchewan, Saskatoon, SK S7N 5E5, Canada; mojtaba.shafiee@usask.ca (M.S.); lok128@mail.usask.ca (K.H.); 2Margaret Ritchie School of Family and Consumer Sciences, University of Idaho, Moscow, ID 83843, USA; vlane@uidaho.edu; 3School of Public Health, University of Saskatchewan, Saskatoon, SK S7N 2Z4, Canada; michael.szafron@usask.ca; 4Department of Community Health and Epidemiology, University of Saskatchewan, Saskatoon, SK S7N 5E5, Canada; pup165@mail.usask.ca; 5Canadian Centre for Health and Safety in Agriculture, University of Saskatchewan, Saskatoon, SK S7N 2Z4, Canada

**Keywords:** food security, traditional foods, COVID-19 pandemic, urban Indigenous peoples, Canada

## Abstract

The COVID-19 pandemic has notably impacted food security, especially among urban Indigenous communities. This study aimed to examine the impact of the pandemic and related lockdown measures on the food security of urban Indigenous peoples in Saskatchewan, Canada. In partnership with Indigenous co-researchers, we designed an online survey disseminated via SurveyMonkey^®^ (San Mateo, CA, USA) from August 2021 to August 2022. This survey detailed background information, the Household Food Security Survey Module (HFSSM), state of food access, and traditional food consumption habits. Of the 130 Indigenous respondents, 75.8% were female, 21.9% male, and 2.3% non-binary, with an average age of 36.2 years. A significant 68.4% experienced food insecurity during the pandemic’s first four months. Increased food prices (47.1%) and reduced market availability (41.4%) were the dominant causes. Additionally, 41.8% highlighted challenges in accessing traditional foods. Relying on community resources and government food distribution programs (40.7%) was the most reported coping strategy for those experiencing food insecurity. Notably, 43.6% reported receiving no government financial support during the crisis. This study emphasizes the severe food insecurity among urban Indigenous communities in Saskatchewan during the pandemic. The findings highlight the immediate need for interventions and policies that ensure access to culturally relevant food, especially for future crises.

## 1. Introduction

Before the onset of the coronavirus disease (COVID-19) pandemic, food insecurity was already a pressing public health concern in Canada, with 12.7% of households experiencing inadequate or insecure access to food due to financial constraints [1]. Of particular concern was the disproportionate impact of food insecurity on off-reserve Indigenous households, where the food insecurity rate was as high as 28.2% [1]. Our earlier research, focused on the pre-pandemic era, showed that all pillars of food security, including availability, accessibility, utilization, and stability, were affected among Indigenous peoples living on- and off-reserve in Canada [2].

For Indigenous peoples, food security encompasses not only access to conventional Western foods but also to traditional foods which have been a cornerstone of cultural, spiritual, and physical well-being for generations [3]. Traditional foods not only provide unique nutritional benefits but are also closely intertwined with cultural identity, community ties, and ancestral knowledge [4]. The four pillars of food security take on additional depth when applied to traditional foods: the availability of traditional hunting, fishing, and gathering grounds; the accessibility of these areas and the knowledge to harvest food sustainably [5]; the utilization of traditional foods in diets and ceremonies; and the stability of these food sources in the face of environmental and societal changes [6]. It is essential to understand that food security for Indigenous communities involves a dual focus—both on Western foods and on maintaining and revitalizing traditional food systems [7].

The COVID-19 pandemic has had a profound impact on global health and the economy, leading to the implementation of measures like stay-at-home orders, social distancing, travel restrictions, and business closures to curb its spread [8]. These measures have disrupted several aspects of life, including food access and security, and vulnerable populations like urban Indigenous peoples are particularly susceptible to the pandemic’s negative impacts due to pre-existing systemic inequalities [9,10]. Disruptions in food access and availability can detrimentally affect the physical, mental, and social well-being of urban Indigenous peoples, as the link between household food security and individuals’ health and well-being is well established [7,11,12,13,14]. The COVID-19 pandemic has brought about changes in the depth and breadth of food insecurity, which could have serious and long-lasting health consequences [15].

A few studies have investigated the effects of the COVID-19 pandemic on food insecurity among Indigenous populations worldwide [16,17,18]. For instance, McAuliffe and colleagues conducted an online survey with a nationally representative sample of Canadian adults in May 2020. The results indicated that 17.3% of the participants reported concerns about having enough food due to the pandemic, with a higher prevalence among those who identified as Indigenous (27.1%) [16]. John-Henderson et al. conducted a longitudinal survey involving American Indian adults living on the Blackfeet reservation in northwest Montana during the pandemic. The study spanned four months (24 August 2020–30 November 2020), and the findings revealed that by the end of the survey, 79% of participants reported a significant increase in food insecurity [17]. Arriagada et al. expressed concerns about the vulnerability of Indigenous peoples in urban areas of Canada due to the socioeconomic impacts of COVID-19, considering that more than one in three Indigenous individuals in urban areas lived in food-insecure households in 2017 [19]. While these studies shed light on the broader issue, there remains a noticeable gap in the literature regarding the specific circumstances of urban Indigenous peoples in Saskatchewan. This underscores the need for further research in this area.

Understanding the immediate implications of the pandemic on the food security of Indigenous peoples residing in urban areas of Canada is not only necessary for providing immediate assistance to this vulnerable population, but also for developing policies and programs that can better prepare us for future public health and economic crises. With this in mind, the aim of this study is to examine the impact of the COVID-19 pandemic and related lockdown measures on the food security of urban Indigenous peoples in Saskatchewan, Canada. By conducting this research, we seek to contribute valuable insights to the existing knowledge and inform evidence-based strategies that can effectively support urban Indigenous communities during times of crisis.

## 2. Materials and Methods

### 2.1. Data Collection and Study Population

In this cross-sectional study, we developed an online semi-structured survey questionnaire consisting of four subscales: (1) background information (12 questions), (2) household food security status (18 questions), (3) traditional food consumption (7 questions), and (4) the state of food access (6 questions). The survey questionnaire was constructed by integrating multiple sources. Initially, the foundation was established using the 18-item Household Food Security Survey Module (HFSSM). This was further enriched through consultations with Canadian experts specializing in food security and in close communication with Indigenous co-researchers. Additionally, we drew on insights from the existing literature, with particular emphasis on an online survey executed in two COVID-19-affected cities: Wuhan and Nanjing, in China [20,21]. To ensure the cultural appropriateness and relevance of the questionnaire, 10 Indigenous co-researchers were recruited to provide comments and feedback on the survey between January 2021 and June 2021. Co-researchers were given the option to provide feedback via video conferencing, phone call, or email, based on their availability and preference. Following the co-researchers’ feedback, revisions were made to the survey questionnaire, and it was then distributed to Indigenous adults living in urban areas of Saskatchewan in August 2021 using SurveyMonkey^®^ (San Mateo, CA, USA).

The sample size for this study was driven by feasibility and our collaborative approach. Our consultations with Indigenous co-researchers provided us with insights regarding anticipated community interest and outreach. Their guidance and the inclusive nature of this research directed us to garner as diverse a response as feasible within our time constraints. Consequently, the survey was kept open from August 2021 to August 2022, and by its conclusion, we garnered insights from 130 urban Indigenous individuals, including 102 First Nations and 28 Métis.

To recruit participants for the study, a recruitment message was sent via email to our Indigenous partner organization, Network Environments for Indigenous Health Research (NEIHR), which was subsequently forwarded to urban Indigenous peoples. The message contained a link to the online survey and provided details about the purpose of the study, the inclusion criteria, and the researchers’ contact information. Prior to starting the survey, participants were presented with a page containing study details, including confidentiality and their right to discontinue at any time. Participants gave informed consent before proceeding to the survey questions. To incentivize participation, participants were entered into a draw to win one of four available gift cards. Email addresses were captured separately from the survey data to ensure the anonymity of the responses. This research project was approved by the University of Saskatchewan Behavioral Research Ethics Board (Beh 2162).

### 2.2. Background Information

We asked participants about their age, gender, Indigenous identity, marital status, household size, education level (both individual and household), total household income, home ownership, employment status, and health insurance. We also asked if they received government financial support during the COVID-19 crisis.

### 2.3. Household Food Security Status

Household food security status during the initial 4 months of the COVID-19 pandemic was measured using HFSSM, Canada’s primary validated measure of food security [22,23]. While the standard approach is to examine the past 12 months, we focused on this shorter timeframe to investigate the pandemic’s impact on food security. HFSSM specifically focuses on economic difficulties in accessing certain, sufficient, or adequate food. It includes 18 questions that range in severity from “worrying about running out of food” to “children going the whole day without food”. Ten of the questions are specific to the experiences of adults, while the remaining eight are specific to the experiences of children under 18 years of age in the household. We used the HFSSM to determine the food security status of households, with adult food security status being considered as the household food security status in households without children. For households with children under the age of 18, we analyzed both the adult and child scales to determine their food security status. We classified households into different levels of food security based on the number of affirmative responses.

### 2.4. State of Food Access

Participants were asked about their food access during the COVID-19 pandemic, including the frequency of physical and online store purchases, challenges accessing food, and changes in food expenses. Participants were also asked how they coped with the challenges they faced related to food access. These questions aimed to gather a comprehensive understanding of the challenges and experiences participants faced regarding food access during the COVID-19 pandemic.

### 2.5. Traditional Food Consumption

The HFSSM does not account for the importance of traditional food practices in Indigenous peoples’ cultural identity and health [3,24]. To address this gap, we included seven multiple-choice and open-ended questions to understand participants’ traditional food consumption habits and attitudes, as well as factors that may prevent individuals from eating traditional foods and limitations on the availability and accessibility of these foods. Additionally, we asked if participants’ households experienced any challenges accessing traditional foods during the COVID-19 pandemic.

### 2.6. Statistical Analysis

Descriptive statistics were used to analyze the data collected in this study. Frequencies and percentages were calculated to summarize categorical variables such as participants’ demographic information. For the continuous variable, age, we tested its normality of distribution using the Kolmogorov–Smirnov test. Age exhibited a normal distribution, which led us to compute its mean values and standard deviations. For open-ended qualitative data, a thematic analysis approach was adopted. This involved coding the responses, grouping them into themes, and interpreting the patterns observed. All statistical and qualitative analyses were performed using SPSS statistical package, version 28.

## 3. Results

### 3.1. Background Information

As reported in Table 1, the survey was completed by a total of 130 respondents, of which 78.5% identified as First Nations and 21.5% identified as Métis. The mean age of the participants was 36.2 ± 12.5 years. When looking at gender distribution, 75.8% identified as female, 21.9% as male, and a small minority identified as other or chose not to specify. Educationally, over a third (34.1%) of the respondents had pursued higher education, holding either a bachelor’s degree or an advanced degree. In contrast, 27.0% had attained a high school diploma or its equivalent. Financially, a notable portion (42%) reported an annual household income ranging from CAD 0 to CAD 39,999. Employment-wise, just under half (49.6%) were in full-time positions. For a comprehensive breakdown of the marital status, household dynamics, housing situations, and more nuanced employment details, readers are directed to Table 1.

As presented in Table 1, most respondents (48.0%) reported having federal health insurance through their status as an Indigenous person, with 34.6% reporting having health insurance through work. Finally, just over half of the participants received financial support from the government during the COVID-19 crisis, with the most common form being the Canada Emergency Response Benefit (CERB) (29.4%).

### 3.2. Household Food Security Status

Figure 1 reveals that 68.4% of urban Indigenous individuals in Saskatchewan experienced some level of food insecurity during the first four months of the COVID-19 pandemic. Among them, 33.3% experienced severe food insecurity, 22.8% experienced moderate food insecurity, and 12.3% experienced marginal food insecurity.

### 3.3. State of Food Access

Table 2 details households’ food access patterns during the first four months of the COVID-19 pandemic. Most households predominantly shopped in physical stores less frequently, with a substantial number not utilizing online stores for food purchases. Major challenges faced included increasing food prices, limited availability at supermarkets, restricted access to these supermarkets, and pandemic-related income losses. Comparatively, food expenses increased for many households after the onset of COVID-19. The pandemic also ushered in food security issues, with households unable to eat preferred foods, facing limited food variety, and some experiencing outright food shortages.

Table 3 indicates that 54 respondents answered the open-ended question, “What did you do to cope with the challenges you faced related to access to food?”. The responses were grouped into eight different categories. The most common coping strategy reported by the respondents was relying on community resources and government food distribution programs, such as food banks and low-cost community markets (40.7%), followed by changing eating habits (33.3%), turning to family and friends for financial and food support (27.8%), and adopting budgeting and meal planning strategies (24.1%). A smaller proportion of participants coped by gardening (9.3%), looking for sales and deals (7.4%), seeking out alternative sources of protein (5.6%), and prioritizing their children’s food needs (5.6%) (Table 3).

### 3.4. Traditional Food Consumption

Our findings highlight the varied consumption patterns of Indigenous traditional foods among participants. While a majority rarely or never partake in traditional foods, a minority consume them daily (Table 4). Most participants lean towards store-bought foods, though a notable percentage maintain a balance between store-bought and traditional foods.

Various factors hinder participants from consuming traditional foods. Time constraints, accessibility issues, and a lack of knowledge regarding preparation stand out as primary challenges (Table 4). Additionally, barriers to traditional food availability primarily encompass land access difficulties and gaps in traditional food preparation knowledge. Climate change emerges as an external factor influencing food availability. The COVID-19 pandemic introduced further complications, with many households facing obstacles in accessing traditional foods during this period.

Two open-ended questions, “Can you please list the Indigenous traditional foods you eat most often?” and “What do you think are the most important advantages of Indigenous traditional foods?” were asked. Participant responses to these two open-ended questions generated several emergent themes (Table 3). A total of 73 participants answered the open-ended question about listing the Indigenous traditional foods they eat most often. The most common categories were wild/game meat (80.8%), fish (50.7%), and wild plants and berries (45.2%). Other categories mentioned less frequently included breads and pastries (e.g., bannock) (35.6%), soups and stews (e.g., bone broth) (5.5%), and fermented foods (i.e., sauerkraut) (2.7%). Among those who reported consuming wild/game meat, moose (66.1%) was the most frequently consumed type, followed by deer (45.8%), and elk (13.6%). A total of 65 respondents provided information on what they believe to be the most important advantages of Indigenous traditional foods. The responses were grouped into six different categories based on their similarity (Table 3). The most commonly cited advantage of Indigenous traditional foods was health and nutrition, with 64.6% of respondents highlighting the nutrient density, lack of additives and preservatives, and overall health benefits of these foods. Cultural significance was also a significant theme, with 35.4% of respondents emphasizing the connection of food to the land, the ethical harvesting and processing of these foods, and their role in intergenerational knowledge transfer and community building. Taste and enjoyment, affordability and accessibility, environmental sustainability, and digestibility and shelf life were also mentioned by participants, albeit to a lesser extent.

## 4. Discussion

The COVID-19 pandemic has had a significant impact on food security among urban Indigenous individuals in Saskatchewan, Canada. Our survey found that 68.4% of respondents experienced some level of food insecurity during the first four months of the pandemic, with 33.3% experiencing severe food insecurity. The most common food access challenges experienced by households during the pandemic were food price increases and limited food availability at food markets and supermarkets. Coping strategies included relying on community resources and government food distribution programs, changing eating habits, and turning to family and friends for financial and food support.

Before the pandemic, data from the 2017–2018 Canadian Community Health Survey (CCHS) showed that 28.2% of off-reserve Indigenous peoples experienced food insecurity [1]. Our finding that Indigenous peoples living in urban areas of Saskatchewan experienced high levels of food insecurity during the first few months of the pandemic is supported by previous research [16,17,18,25,26]. McAuliffe et al. conducted an online survey in May 2020 with a nationally representative sample of Canadian adults, and their findings showed that 17.3% of the participants reported food worry due to the pandemic, with a higher prevalence among those identifying as Indigenous (27.1%) [16]. Similarly, an online survey released in March 2020 showed that more than 77% of Native American respondents experienced food insecurity during the first 3 months of the pandemic, with 31.1% experiencing severe food insecurity [25]. The results of another survey conducted in New Mexico in May and June 2020 demonstrated that more than 46% of Native American respondents were classified as food insecure and reported high degrees of food-related worries and concern due to the coronavirus outbreak [26]. A longitudinal survey involving American Indian adults on the Blackfeet reservation in northwest Montana revealed that 79% of participants reported a significant increase in food insecurity over a four-month period [17]. In a cross-sectional survey conducted in Tasmania, Australia, during the pandemic (late May to early June 2020), it was found that the prevalence of food insecurity was 26%. However, individuals who identified as Aboriginal and/or Torres Strait Islander had over three times higher odds of experiencing food insecurity [18]. These studies, alongside the findings from our study, provide additional evidence of the heightened food insecurity experienced by Indigenous populations during the pandemic. The heightened severity of food insecurity underscores the compounded vulnerabilities that these communities face. This could be attributed to specific socioeconomic and cultural contexts, which necessitate tailored interventions.

Food price increases and limited food availability at markets and supermarkets were the most common food access challenges experienced by households. Research indicates that the global food supply chains experienced disruptions during the COVID-19 pandemic, leading to fluctuations in food prices [27,28,29,30,31,32]. Additionally, in the early stages of the pandemic, there was a significant disruption in food availability. Grocery store visitors encountered empty shelves as the food supply chain struggled to meet the heightened demand caused by consumer stockpiling, changes in demand from restaurants and foodservice establishments, and workforce illnesses [26]. During the pandemic, Indigenous community members in Canada who engaged in virtual talking circles expressed concerns regarding the rising expenses associated with accessing food [33]. The escalation in prices can be traced back to the economic aftermath of the pandemic, including heightened transportation limitations resulting from lockdown measures. In New South Wales, Indigenous community members reported heightened food insecurity for certain groups due to fear of entering crowded shopping centers and the perceived risk of contracting COVID-19. Local shops were also accused of inflating prices, exacerbating the challenges faced by the community [34]. Similarly, an investigation examining the effects of COVID-19 on food supply and purchasing patterns in a rural Australian supermarket revealed the various challenges faced by retailers. These challenges included empty shelves influenced by media and government communication, product unavailability, and community fear [35]. Customers reported concerns about contracting COVID-19, the unavailability of food, rising prices, and government-imposed restrictions, all of which influenced their purchasing behavior. Overall, the COVID-19 pandemic has posed significant challenges to food accessibility, including price increases, limited availability, and supply chain disruptions.

The most frequently utilized coping strategies during the pandemic included accessing community resources and government food distribution programs, changing eating habits, and seeking financial and food support from family and friends. Similar results from a cross-sectional multi-country online survey conducted between May and July 2020, including in Canada, highlighted that 32.7% of participants identified price increases as a significant obstacle to acquiring food, resulting in the decreased variety (50.4%), quality (30.2%), and quantity (39.2%) of available food options [36]. To mitigate the impact of restrictive public health measures on their diets, individuals employed alternative strategies, including relying on less-preferred foods (49.2%), reducing portion sizes (30.3%), seeking food assistance from others (17.0%), and accessing food aid programs (8.3%). According to an online survey, 62% of respondents from Australia, Canada, Mexico, the United Kingdom, and the United States reported reduced consumption of food prepared outside their homes as a result of the pandemic [37]. Similarly, Indigenous community members in Canada who participated in virtual talking circles expressed a shift towards self-reliance by relying less on outside sources for food. They highlighted the importance of growing their own produce and establishing their own food market as strategies to enhance food security [33], which align with our study findings. Moreover, Indigenous community members from New South Wales expressed concerns about the rising prices set by local shops, leading them to resort to purchasing cheaper but potentially less nutritious options to feed their families [34]. Data collected from Indigenous communities and local populations in South America, specifically Colombia, Ecuador, Guyana, and Peru, demonstrated that wildlife consumption increased as a short-term response to food insecurity during the pandemic. However, the rise in wild meat consumption was hindered by high prices and limited availability due to unsuccessful hunts [38]. These findings highlight the diverse coping strategies employed by individuals and communities to address food access challenges during the pandemic. The shift towards self-reliance and the emphasis on community-centric solutions offer insights into the resilience and adaptive capacities of urban Indigenous communities.

Strengths and Limitations: One strength of this study is the partnership with Indigenous co-researchers, ensuring cultural sensitivity and an inclusive approach to data collection. Additionally, the use of an online survey allowed for a wide geographic reach and convenient participation. However, the study’s reliance on self-reported data introduces potential biases and recall errors. The online nature of the survey may limit participation from individuals who do not have access to the internet or technology, potentially excluding voices from marginalized populations. Furthermore, our sample was predominantly from urban Saskatoon, which raises questions about the generalizability of our findings to other urban Indigenous communities across Canada. However, it provides insight on the experiences and challenges faced by Indigenous populations in larger metropolitan areas, emphasizing the need for more comprehensive studies in such settings to fully understand the scope of food security issues.

Implications: The findings of this study have important implications for policymakers and stakeholders involved in addressing food security among urban Indigenous populations. The identified barriers, such as food price increases and limited availability, require targeted interventions to ensure equitable access to nutritious food. Strengthened community resources and government food distribution programs can play a crucial role in supporting vulnerable households. Additionally, supporting traditional food practices and addressing challenges to accessing traditional foods are important for cultural preservation and holistic well-being. Long-term policies should focus on building resilient food systems and addressing underlying systemic issues to promote food security among urban Indigenous peoples.

## 5. Conclusions

This study provides valuable insights into how the COVID-19 pandemic impacted the food security of urban Indigenous peoples in Canada. The findings highlight the exacerbated challenges faced by these individuals already facing a high level of food insecurity, such as food price increases, limited availability, and barriers to accessing traditional foods. Immediate policy action is crucial to address these issues and support the food security and well-being of urban Indigenous populations. Moreover, the knowledge gained from this study has broader implications beyond the current pandemic, serving as a foundation to better prepare for future public health and economic crises. By implementing proactive measures, promoting equitable access to affordable and nutritious food, and strengthening community support systems, we can work towards building more resilient and inclusive food systems that prioritize the health and dignity of all individuals and communities.

## Figures and Tables

**Figure 1 nutrients-15-04278-f001:**
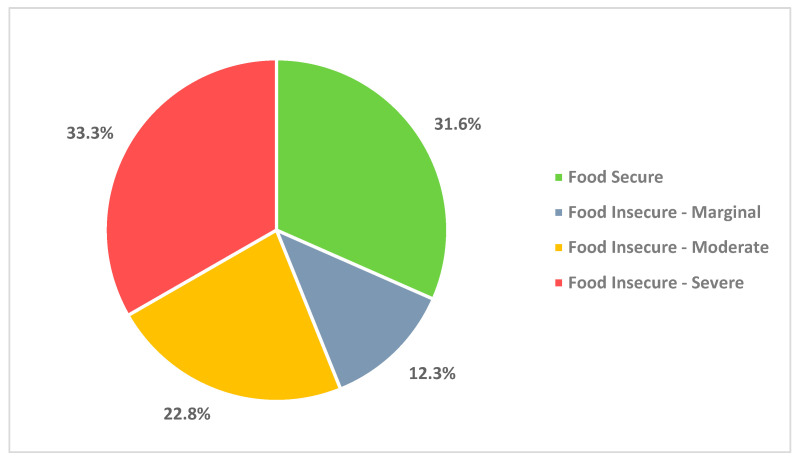
Prevalence of household food insecurity during the first four months of the COVID-19 pandemic.

**Table 1 nutrients-15-04278-t001:** Characteristics of study participants.

Variables/Questions	*n* (%)
Gender identity	
Male	28 (21.9)
Female	97 (75.8)
Prefer not to say	1 (0.8)
Other	2 (1.5)
Missing	2
Indigenous identity	
First Nations (status or non-status)	102 (78.5)
Métis	28 (21.5)
Missing	0
Marital status	
Married	32 (25.6)
Living common-law/long-term relationship	27 (21.6)
Widowed/Divorced/Separated	12 (9.6)
Single	54 (43.2)
Missing	5
Household size	
1 member	18 (14.4)
2 members	23 (18.4)
3 members	30 (24.0)
4 members	30 (24.0)
5 or more members	24 (19.2)
Missing	5
Highest level of formal education	
Less than high school diploma or its equivalent	18 (14.3)
High school diploma or a high school equivalency certificate	34 (27.0)
Certificate/diploma—trade/college/non-university/university below bachelor	31 (24.6)
Bachelor degree or university certificate/diploma/degree above bachelor level	43 (34.1)
Missing	4
Total household income from all sources (CAD) ^1^	
$0 to $19,999	27 (21.3)
$20,000 to $39,999	26 (20.5)
$40,000 to $59,999	21 (16.5)
$60 000 to $79,999	14 (11.0)
$80,000 to $99,999	12 (9.5)
$100,000 and higher	27 (21.2)
Missing	3
Home ownership	
Rent	72 (57.1)
Own	36 (28.6)
Live with family members	16 (12.7)
Shelter	0 (0.0)
Other	2 (1.6)
Missing	4
Employment status	
Full time job	63 (49.6)
Part time job	14 (11.0)
Not working, looking for work	23 (18.1)
Not working, unable to work	21 (16.6)
Household manager	6 (4.7)
Missing	3
Health insurance	
None	9 (7.1)
Yes, through work (Employment Insurance)	44 (34.6)
Yes, through status (Indian Status)	61 (48.0)
Yes, through the province (Provincial Insurance)	4 (3.2)
Yes, other	9 (7.1)
Missing	3
Received government financial support during COVID-19?	
Canada Emergency Response Benefit (CERB ^2^)	37 (29.4)
Canada Emergency Student Benefit (CESB ^3^)	11 (8.7)
Unemployment benefits	10 (7.9)
Temporary lay-off/maternity/Employment Insurance (EI)	4 (3.2)
Pension benefits	3 (2.4)
Saskatchewan Assistance Program (SAP ^4^)	16 (12.7)
Other	11 (8.7)
None of the above	55 (43.6)

^1^ CAD 0 to 19,999 = USD 0 to 14,999.25, CAD 20,000 to 39,999 = USD 15,000 to 29,999, CAD 40,000 to 59,999 = USD 30,000 to 44,999, CAD 60,000 to 79,999 = USD 45,000 to 59,999, CAD 80,000 to 99,999 = USD 60,000 to 74,999, CAD 100,000 and higher = USD 75,000 and higher. ^2^ A financial support provided to employed and self-employed individuals directly affected by COVID-19. ^3^ Designed specifically for post-secondary students and recent graduates who could not find employment due to the pandemic. ^4^ A provincial program designed for Saskatchewan residents that provides financial aid to individuals and families.

**Table 2 nutrients-15-04278-t002:** State of food access.

Question	*n*	%
During the first four months of COVID-19 pandemic, how often did your household buy food in physical stores (such as farmer’s markets, supermarkets, and convenient stores/gas stations)?		
Never	7	6.7
Less than a time per week	63	60
2–3 times per week	29	27.6
4–6 times per week	5	4.7
7 times per week	1	1.0
During the first four months of COVID-19 pandemic, how often did your household buy food from online stores?		
Never	68	65.4
Less than a time per week	26	25.0
2–3 times per week	8	7.7
4–6 times per week	2	1.9
7 times per week	0	0.0
During the first four months of COVID-19 pandemic, did your household experience any challenges accessing food? Check all that apply.		
Food price increase	49	47.1
Limited food availability at food markets and supermarkets (e.g., food out of stock)	43	41.4
Restricted access to food markets and supermarkets	33	31.7
Restricted mobility to leave the home	31	29.8
Loss of income due to COVID-19 outbreak	26	25.0
Access to Indigenous/ First Nations’ foods (wild meat, fish, etc.)	19	18.3
Restricted food delivery services to your home	16	15.4
Restricted access to online stores (e.g., restricted access to Wi-Fi and technologies)	14	13.5
Limited food availability at online stores (e.g., food out of stock)	8	7.7
Other	4	3.9
None of the above	18	17.3
Comparing to food expenses before the COVID-19 outbreak, how much more/less is your household spending on food?		
Less than before	8	7.7
About the same as before	29	27.9
A little more than before (up to twice as much)	47	45.2
Much more than before (more than double)	19	18.3
Did your household experience the following food security challenges due to the COVID-19 outbreak? Check all that apply.		
Not able to eat the kinds of foods preferred	50	51.0
Having to eat a limited variety of foods	48	49.0
Having to eat some foods that you really did not want to eat	34	34.7
Not having enough food	28	28.6
Having to eat a smaller meal than you felt you needed because there was not enough food	24	24.5
Having to eat fewer meals in a day because there was not enough food	19	19.4
Low access to food hampers	19	19.4
Information pertaining to bulk buying, freezing and other food security measures	18	18.4
Information pertaining to how to eat healthy on a budget	17	17.4
Going to sleep at night hungry because there was not enough food	9	9.2
Going a whole day and night without eating anything because there was not enough food	7	7.1
There was no food to eat of any kind	5	5.1
Other	2	2.0
None of the above	27	27.5

**Table 3 nutrients-15-04278-t003:** Emergent themes from responses to open-ended questions.

Emergent Theme	Definition/Description	# of Mentions	%
1. What did you do to cope with the challenges you faced related to access to food? (*n* = 54)			
Community resources & government food distribution	Relying on community resources and government food distribution programs, such as community food boxes, food banks, and low-cost community markets	22	40.7
Changing eating habits	Making changes to their eating habits, including eating less, reduced snacking, substituting food, and eating cheaper, less healthy options.	18	33.3
Family and social network support	Turning to family and friends for financial and food support and borrowing money to purchase food.	15	27.8
Budgeting and meal planning	Adopting budgeting and meal planning strategies to manage their food budget effectively.	13	24.1
Gardening	Gardening and storing food to ensure a reliable supply of food.	5	9.3
Seeking out deals and sales	Looking for deals and sales to save money on their food purchases.	4	7.4
Alternative protein sources	Seeking out alternative sources of protein, such as hunting, fishing, and protein powder shakes.	3	5.6
Putting Children First	Making sure their children were fed first before themselves and prioritizing buying food for their kids before getting what they wanted.	3	5.6
2. Can you please list the Indigenous traditional foods you eat most often? (*n* = 73)			
Wild/game meat	Moose, deer, elk, rabbit, duck, wild chicken, dried meat, caribou, bison, pemmican	59	80.8
Fish	Fish (e.g., Salmon)	37	50.7
Wild Plants and Berries	Berries, wild rice, wild mint	33	45.2
Breads and Pastries	Bannock	26	35.6
Soups and Stews	Bullet soup and Bone broth	4	5.5
Fermented Foods	Sauerkraut	2	2.7
3. What do you think are the most important advantages of Indigenous traditional foods (*n* = 65)			
Health and Nutrition	This category includes responses such as “Healthy”, “Nutritious options, rich mineral and vitamin source”, “Leaner and less fat content”, “Not processed”, “No preservatives—No additives”, and “No added hormones or antibiotics”. These responses highlight the health benefits of traditional Indigenous foods, which are often less processed and more nutrient-dense than modern, Western foods.	42	64.6
Cultural Significance	This category includes responses such as “Tradition and Culture”, “Connection to food and land”, “Ethical harvesting and processing”, “Intergenerational transfer of knowledge”, and “Brings peoples together”. These responses highlight the cultural and social significance of Indigenous traditional foods, which are often deeply intertwined with Indigenous cultures and ways of life.	23	35.4
Taste and Enjoyment	This category includes responses such as “Taste” and “Sharing”. These responses highlight the sensory and social aspects of traditional Indigenous foods, which are often enjoyed as part of communal meals and celebrations.	13	20.0
Affordability and Accessibility	This category includes responses such as “Cost/Affordability” and “Accessibility”. These responses highlight the practical benefits of traditional Indigenous foods, which are often more affordable and accessible to Indigenous communities than Western foods.	12	18.5
Environmental Sustainability	This category includes responses such as “Natural and organic”, “Little/no pesticides and herbicides”, and “Less chemicals”. These responses highlight the environmental benefits of traditional Indigenous foods, which are often produced using traditional, sustainable methods that have a smaller ecological footprint than modern industrial agriculture.	7	10.8
Digestibility and Shelf Life	This category includes responses such as “Easier to digest” and “Shelf life”. These responses highlight the practical advantages of traditional Indigenous foods, which are often easier to digest and have a longer shelf life than modern, highly processed foods.	7	10.8

**Table 4 nutrients-15-04278-t004:** Traditional food consumption.

Question	*n*	%
How often do you eat Indigenous traditional foods such as fish, wild meat, berries, etc.?		
Rarely/never	53	53.6
1 per week	22	22.2
2–4 per week	20	20.2
5–6 per week	1	1.0
≥1 per day	3	3.0
What is your Traditional food preference?		
Only store-bought foods and little to no traditional foods	20	20.6
Mostly store-bought foods and some traditional foods	40	41.2
About the same amount of traditional and store-bought foods	21	21.7
Mostly traditional foods and some store-bought foods	13	13.4
Only traditional foods and little to no store-bought foods	3	3.1
What factors prevent you from eating traditional foods? Check all that apply.		
No time to hunt, fish, forage, or prepare	52	54.2
No access/availability	36	37.5
Lack of knowledge of how to cook with it, prepare it, or access to it	36	37.5
Do not have kinship ties with those who hunt, fish, or forage.	28	29.2
Expense to buy bullets, fishing gear, etc.	22	22.9
Can’t get on land	22	22.9
Preference/lifestyle	19	19.8
Don’t like it	6	6.3
Unhealthy	6	6.3
Other	14	14.6
What factors limit the traditional food availability/accessibility? Check all that apply.		
Lack of access to the land	49	54.4
Lack of traditional knowledge to harvest and prepare traditional foods	44	48.9
Climate change affecting traditional food harvesting and consumption (e.g., fires, sea-level rise, flooding, drought, or animal disease)	35	38.9
Financial costs for acquiring traditional foods	33	36.7
Geographic isolation (i.e., living in isolated and remote regions such as La Loche)	9	10.0
Other	12	13.3
Did your household experience any challenges accessing traditional foods due to the COVID-19 outbreak?		
Yes	41	41.8
No	32	32.7
Don’t know	22	22.4
Refused	3	3.1

## Data Availability

The data presented in this study are available on request from the corresponding author. The data are not publicly available due to privacy and ethical considerations related to working with Indigenous communities.

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
