# Peer review of "Exploring the Implications of COVID-19 on Food Security and Coping Strategies among Urban Indigenous Peoples in Saskatchewan, Canada"

_nutrients, 2023, doi:10.3390/nu15194278_

Round 1

Reviewer 1 Report

·        The manuscript addresses an important and timely issue – the impact of the COVID-19 pandemic on food security among urban Indigenous communities. This is a topic of considerable societal concern and relevance.

·        The use of an online survey, in partnership with Indigenous co-researchers, enhances the study's credibility and cultural sensitivity. The inclusion of the Household Food Security Survey Module (HFSSM) strengthens the assessment of food security.

·        The manuscript provides a thorough analysis of the survey results, including demographic information, the prevalence of food insecurity, causes, coping strategies, and government support. The breakdown of demographic data enriches the understanding of the sample.

·        The manuscript effectively highlights the urgent need for interventions and policies to ensure access to nutritious, culturally relevant food for Indigenous communities during crises such as the COVID-19 pandemic.

Author Response

Reviewer 1:

The manuscript addresses an important and timely issue – the impact of the COVID-19 pandemic on food security among urban Indigenous communities. This is a topic of considerable societal concern and relevance.

The use of an online survey, in partnership with Indigenous co-researchers, enhances the study's credibility and cultural sensitivity. The inclusion of the Household Food Security Survey Module (HFSSM) strengthens the assessment of food security.

The manuscript provides a thorough analysis of the survey results, including demographic information, the prevalence of food insecurity, causes, coping strategies, and government support. The breakdown of demographic data enriches the understanding of the sample.

The manuscript effectively highlights the urgent need for interventions and policies to ensure access to nutritious, culturally relevant food for Indigenous communities during crises such as the COVID-19 pandemic.

Response: Thank you for your thoughtful feedback on our manuscript. We are grateful for your time and thorough review.

Reviewer 2 Report

The manuscript entitled “Exploring the Implications of COVID-19 on Food Security and Coping Strategies among Urban Indigenous Peoples in Saskatchewan, Canada” presents interesting issue, however some corrections are needed.

Introduction:

- I have no comments

Materials and Methods:

- Lines 93-98 – How were the questionnaire questions developed?

- Lines 127 – “Household Food Security Survey Module (HFSSM) “ Is this questionnaire validated? Is its reliability measured? 

- More information about questionnaire is needed.

- Line 150 – “seven multiple-choice and open-ended questions..” – Please add information about the basis on which these questions were created, where they were sourced from?.

- Line 158-159 – “Mean values and standard deviations were computed for continuous variables such as age” - Was the normality of distribution tested? The information about it should be added and authors should be consequent. If data have normal distribution, they should be treated as such, if not, nonparametric tests should be applied.  Please specify it.

- How was the sample size established?

Results 

- Table 1. Values is presented in dollars or Canadian dollars? Please specified it. Perhaps presenting the values in an international currency would be a good idea

- Table 1 - During the COVID-19 crisis, have you received any of the following government financial supports?  Canada Emergency Response Benefit (CERB) - "International readers don't know much about Canadian government financial supports?  This needs to be explained.

- A summary table of the results would be helpful – not just providing the percentage of responses to individual questions. It may bore the reader."

Discussion: 

- I have no comments

Author Response

Reviewer 2:

The manuscript entitled “Exploring the Implications of COVID-19 on Food Security and Coping Strategies among Urban Indigenous Peoples in Saskatchewan, Canada” presents interesting issue, however some corrections are needed.

Introduction:

- I have no comments

Materials and Methods:

- Lines 93-98 – How were the questionnaire questions developed?

Response: We appreciate the reviewer's insightful comments on our manuscript. The survey questionnaire was constructed by integrating multiple sources. Initially, the foundation was established using the 18-item Household Food Security Survey Module (HFSSM). This was further enriched through consultations with Canadian experts specializing in food security and in close communication with Indigenous co-researchers. Additionally, we drew insights from the existing literature, with particular emphasis on an online survey executed in two COVID-19-affected cities: Wuhan and Nanjing, in China. This information was added to the manuscript. Page 3, Line 98-103.

- Lines 127 – “Household Food Security Survey Module (HFSSM) “ Is this questionnaire validated? Is its reliability measured?

Response: Yes, according to Statistics Canada, the Household Food Security Survey Module (HFSSM) is Canada's primary validated measure of food security. Page 3, Line 138. This tool has been used for assessing the food security status of off-reserve Indigenous people since 2004 through national surveys.

- More information about questionnaire is needed.

Response: We added more information about the questionnaire. Page 3, Line 98-103.

- Line 150 – “seven multiple-choice and open-ended questions..” – Please add information about the basis on which these questions were created, where they were sourced from?.

Response: We added more information about the questionnaire. Page 3, Line 98-103.

- Line 158-159 – “Mean values and standard deviations were computed for continuous variables such as age” - Was the normality of distribution tested? The information about it should be added and authors should be consequent. If data have normal distribution, they should be treated as such, if not, nonparametric tests should be applied.  Please specify it.

Response: We indeed tested the normality of the distribution for age as a continuous variable. The age variable demonstrated a normal distribution, which is why we used the mean values and standard deviations in our analysis. Page 4, Line 169.

- How was the sample size established?

Response: The sample size for this cross-sectional study was not pre-determined based on statistical calculations but was rather a product of feasibility and the collaborative approach adopted in this study. Our consultations with Indigenous co-researchers informed us about potential reach and interest within the community. Based on their feedback and the collaborative approach of the study, the aim was to gather as many responses as possible within the given time frame, ensuring robustness in the data collected. The recruitment process was open for a year, and we were able to secure participation from 130 urban Indigenous peoples. Page 3, Line 112-117.

Results

- Table 1. Values is presented in dollars or Canadian dollars? Please specified it. Perhaps presenting the values in an international currency would be a good idea.

Response: The values presented in Table 1 are in Canadian Dollars (CAD). We appreciate the suggestion of converting these values into an international currency for broader comprehension. For the sake of clarity, we have updated Table 1 to specify that the values are presented in Canadian Dollars (CAD). We added the conversion rate to the US dollar to the footnote.

- Table 1 - During the COVID-19 crisis, have you received any of the following government financial supports?  Canada Emergency Response Benefit (CERB) - "International readers don't know much about Canadian government financial supports?  This needs to be explained.

Response: Thank you for your feedback. To assist international readers in understanding the specific Canadian government financial supports mentioned, we have provided detailed explanations in the footnote of Table 1.

- A summary table of the results would be helpful – not just providing the percentage of responses to individual questions. It may bore the reader."

Response: Thank you for the valuable feedback. We recognize the importance of providing concise and clear information to our readers. The decision to present the detailed percentages for individual questions in the table was made to provide a comprehensive overview of the participants' responses, which we believe offers specific insights crucial to the context of the study. However, we'll ensure to incorporate a narrative that highlights the key findings from the table in the text, offering a quick overview while still preserving the depth of information the table provides.

Discussion:

- I have no comments

Reviewer 3 Report

A brief summary

The Introduction and Methods are well explained, with enough information on the given topic. There are many results and it is potentially a good article. It is an interesting topic which requires further research and solutions.

General comments:

In the Results, data from the Tables are repeated in the text, which is redundant, so it is suggested to the authors to decide what to put in the Tables and what is better to leave described in the text. Authors are also suggested to check if all journals are properly listed in the References.

Specific comments:

Line 17  Please add sentence about aim of the study

Line 29-32            In this sentence, please add how this is an important problem in general, and especially in upcoming crises (or similar).

Line 68  Perhaps a more appropriate term would be "A few studies" instead of “Numerous studies” ... given that the authors cite 3 references for this fact.

Line 166                It is suggested to authors to remove “(Table 1) after the first sentence. Everything mentioned after, is more or less also listed in that Table, so you can start this paragraph with: “The Table 1 shows...”  or something similar

Line 181                In Table 1, it is suggested to change the whole question “During the COVID-19 crisis, have you received any of the following government financial supports?” into something simpler, for example: “Government financial support in COVID-19” or similar.

Line 192                The same comment as for Line 166

Line 208                The same comment as for Lines 166 and 192

Line 220                The same comment as for Lines 166, 192 and 208

Line 256                It is suggested to the authors, considering the wide range of the results, to comment a little more in the Discussion. Most of the Discussion section is based on other references.

Line 258-264       This part actually belongs to the Results section

Line 287-288       Is this sentence proven by this study or is it supported by references 24-29? Please, clarify for better understanding.

Line 296                Please delete reference 30 as it is listed after the next sentence.

Line 306                Please delete reference 32 as it is listed after the next sentence.

Line 317                Please delete reference 33 as it is listed after the next sentence.

Line 349                Implications could be included in the Conclusion, but if the authors think that this is better, they can leave it here.

Author Response

Reviewer 3:

A brief summary: The Introduction and Methods are well explained, with enough information on the given topic. There are many results and it is potentially a good article. It is an interesting topic which requires further research and solutions.

General comments:

- In the Results, data from the Tables are repeated in the text, which is redundant, so it is suggested to the authors to decide what to put in the Tables and what is better to leave described in the text. Authors are also suggested to check if all journals are properly listed in the References.

Response: We appreciate the reviewer's insightful comments on our manuscript. In our effort to provide a concise overview of our findings, we have reported only the most significant results from the tables in the text. Our intention was to offer readers a quick snapshot while still preserving the depth of information the tables provide. However, we understand the importance of a streamlined presentation, and we revisited this section to ensure the balance is maintained between clarity and comprehensiveness. As for the References, we conducted a thorough check to ensure all journals are accurately listed and appropriately cited.

Specific comments:

- Line 17, Please add sentence about aim of the study

Response: Thanks, we added the aim of the study to the abstract. Page 1, Line 18.

- Line 29-32, In this sentence, please add how this is an important problem in general, and especially in upcoming crises (or similar).

Response: Thanks, we added this statement to the abstract. Page 1, Line 31.

- Line 68, Perhaps a more appropriate term would be "A few studies" instead of “Numerous studies” ... given that the authors cite 3 references for this fact.

Response: Revised, thanks.

- Line 166, It is suggested to authors to remove “(Table 1) after the first sentence. Everything mentioned after, is more or less also listed in that Table, so you can start this paragraph with: “The Table 1 shows...”  or something similar.

Response: Revised, thanks.

- Line 181, In Table 1, it is suggested to change the whole question “During the COVID-19 crisis, have you received any of the following government financial supports?” into something simpler, for example: “Government financial support in COVID-19” or similar.

Response: Revised, thanks.

- Line 192, The same comment as for Line 166

Response: Revised, thanks.

- Line 208, The same comment as for Lines 166 and 192

Response: Revised, thanks.

- Line 220, The same comment as for Lines 166, 192 and 208

Response: Revised, thanks.

- Line 256, It is suggested to the authors, considering the wide range of the results, to comment a little more in the Discussion. Most of the Discussion section is based on other references.

Response: Thank you for pointing out the need to further elaborate on our results in the Discussion section. We appreciate the feedback and have incorporated additional comments that delve deeper into our findings, ensuring a richer interpretation of our results in the context of existing literature.

- Line 258-264, This part actually belongs to the Results section

Response: We included a brief summary of the main findings at the beginning of the Discussion section to provide context and a foundation for the subsequent discussion.

- Line 287-288, Is this sentence proven by this study or is it supported by references 24-29? Please, clarify for better understanding.

Response: The statement is corroborated by both the cited references and the findings from our study. Page 13, Line 302.

- Line 296, Please delete reference 30 as it is listed after the next sentence.

Response: Removed, thanks.

- Line 306, Please delete reference 32 as it is listed after the next sentence.

Response: Removed, thanks.

- Line 317, Please delete reference 33 as it is listed after the next sentence.

Response: Removed, thanks.

- Line 349, Implications could be included in the Conclusion, but if the authors think that this is better, they can leave it here.

Response: We appreciate the suggestion. Given the significance and importance of the implications for our study, we chose to dedicate a separate paragraph to them, ensuring they are highlighted appropriately. We believe this structure provides clarity and emphasis to the implications of our findings.
